# Immune Functions Alterations Due to Racing Stress in Thoroughbred Horses

**DOI:** 10.3390/ani12091203

**Published:** 2022-05-07

**Authors:** Mônica Cristina Zandoná Meleiro, Hianka Jasmyne Costa de Carvalho, Rafaela Rodrigues Ribeiro, Mônica Duarte da Silva, Cristina Massoco Salles Gomes, Maria Angélica Miglino, Irvênia Luiza de Santis Prada

**Affiliations:** Department of Surgery, School of Veterinary Medicine and Animal Science, University of São Paulo, São Paulo 05508-270, Brazil; mczandona@hotmail.com (M.C.Z.M.); hiankacarvalho@usp.br (H.J.C.d.C.); rafaelarodriguesribeiro@usp.br (R.R.R.); monicasilva@usp.br (M.D.d.S.); cmassoco@gmail.com (C.M.S.G.); irvenia@gmail.com (I.L.d.S.P.)

**Keywords:** immunology, cortisol, high intensity exercise, stress, racehorses

## Abstract

**Simple Summary:**

Racehorses are under constant stress when training and during competitions. It is known that high levels of cortisol, the hormone responsible for stress regulation, can impact the immune system. Hence, the importance of assessing the immune cells, blood components and cortisol during different times of athlete horses’ routines, including races. This research verified the impact of training and racing on the immune function of Thoroughbreds’ trough phagocytosis and oxidative neutrophil burst assays, serum cortisol determination, erythrocytes apoptosis, lymphoproliferation, and blood count analyses.

**Abstract:**

Racehorses are constantly exposed to stress. Aiming to verify the state of blood components and cortisol alterations during their routine and after races, phagocytosis and oxidative neutrophil burst assays, serum cortisol determination, erythrocytes apoptosis evaluation, lymphoproliferation assays, and blood count tests were performed in thirty Thoroughbred racehorses, which were divided in two groups. The samples were taken right after races (moment 0 d), during rest periods (−11 d, +1 d, +3 d), and after training (−8, +2, +5). In both groups, the phagocytosis showed a decrease in percentage and intensity immediately after the race when comparing samples collected during rest or training periods. In the mean values of oxidative burst on samples collected immediately after the race, group I animals demonstrated a decrease (524.2 ± 248.9) when compared with those samples collected in other moments. No significant differences were found between the results of different moments regarding the apoptotic cells and lymphoproliferation assays. The mean values of serum cortisol levels were increased immediately after racing. There was an increase in the percentage of neutrophils found immediately after the race. It was possible to conclude that, although a transient reduction was found in the number of neutrophils, the horses’ adaptive function was not affected.

## 1. Introduction

Racing is one of the many hugely popular equestrian sports, with Thoroughbreds being the most used horses, and thus consequently subjected to intensive training to obtain their best performance [1]. Exercises are of extreme importance to the maintenance of both physical and general well-being, with the transposition of leucocytes to peripheral tissues that happens during mild exercise enhancing the immune system [2]. However, even if essential in moderate levels, exercises can induce intense stress response in most animals, including humans [3,4]. Exercise in horses, such as endurance racing, has been correlated with an increase in plasmatic cortisol levels [5,6]. In young Thoroughbred racehorses, for example, immune changes and increased cortisol secretion due to training were also reported [7,8,9,10]. Cortisol is a natural glucocorticoid hormone produced by the adrenal cortex and is secreted under stressful situations. It regulates some immunological cells and their functions (including neutrophils) that are often reduced after schedules with long periods of training [9,10,11,12]. The importance of neutrophils is due to their action in the nonspecific host defense response, performing the phagocytosis of pathogens and cytokines and synthesis of toxic substances, therefore directly influencing the animal’s response to exercises [13].

The immune system’s main function is to reestablish the body’s homeostasis through several defense mechanisms. The innate system is essential to the immediate response against infectious agents and non-infectious stressors [8,14]. The “open window” hypothesis proposes that after high intensity exercises the immune system can be compromised by a leukocyte function decrease, enhancing the development of respiratory disease both in humans and horse athletes, although there are discussions about the accuracy of the theory in horses [2,7,8].

In horses, studies have been conducted to identify the impacts of exercises on specific immune system components; however, their findings and conclusions regarding the impact on immune function can vary [4,7,15,16]. Currently, it is known that high intensity and short span exercises have immunomodulator effects that can promote inflammation responses, oxidative stress, and transient immune disruption, as well as general alterations in the functional capabilities of cells [17,18,19]. Furthermore, hemopoietic abnormalities and oxidative and cellular stress are present in the airways of horses submitted to intense exercise, a finding that can indicate inflammation of the region in animals that practice intense exercise frequently [20].

Due to artificial selection, Thoroughbred horses display both anatomical and physiological adaptations to this exercise, including their blood parameters and muscles, which were able to rapidly remodel the exercise [8]. An alteration observed in their blood parameters is the higher levels of blood coagulability as well as a decreased clotting time which could be related to adaptations to microvascular injury provoked by intense exercises [10].

Considering that stress caused by intense exercise can impact immune functions, it is important to monitor the cortisol levels, the immune cell system, and alterations in blood components due to training and races in athlete horses. Addressing this issue, we aimed to assess these alterations through neutrophils phagocytosis assay, oxidative burst, lymphocyte proliferation, red blood count, blood cortisol evaluation and apoptosis assays after training, racing and during resting periods.

## 2. Material and Methods

### 2.1. Animals

Thirty Thoroughbred racehorses from the Jockey Club São Paulo (JCSP), varying in age between two and six years (Table 1), were divided into two groups: group I—*n* = 15 and group II—*n* = 15, with samples being taken at different moments. This division was done randomly between the horses with the highest probability of racing in the next seven-day period. The horses were kept in individual stables under natural conditions of luminosity, temperature, and humidity. They were submitted to frequent sanitary control, including immunoprophylactic immunization regimens and control of endoparasites. Sample collection on the day of the race was allowed by the Veterinary Assistance Division (VAD) and the Department of Anti-doping Research Control of the JCSP.

### 2.2. Training Module

The horses underwent the traditional training system for Thoroughbred racehorses in Brazil with morning exercises in step, trot, and soft gallops conducted, in distances ranging between 500 and 4000 m. Besides that, the horses also went through a week of intense exercises with gallop driven by a mounted jockey running through the same distance as that of the competitions (an exercise called “strong distance work”) or half that length (called “speed apron”). In this system, rest periods between track exercises are included, and when the distance and speed exercises are done, a longer rest of three days is given to the horses following the races.

### 2.3. Experimental Design

Blood samples were collected at different times. The sample collection protocol was changed in order to respect the JCSP’s politics as well as diminish the period between the low-exercise day and the racing day and increase the period between this same day and the recovering period. In group I, samples were collected eleven days before the day of the race (training moment −11 d); immediately after the race (moment 0 d); one day after the day of the race (resting moment +1 d); two days after the day of the race (training moment +2 d); and three days after the race (resting moment +3 d). In group II, samples were collected eight days before the race day (training moment −8 d); immediately after the race (moment 0 d); one day after the race (resting moment +1 d); and five days after the race (training moment +5 d). For phagocytosis, oxidative burst, and serum cortisol concentration assays, samples from both groups of animals were used at moments −11 d, −8 d, 0 d, +1 d, +2 d, +3 d, and +5 d. Lymphoproliferation, lymphocyte apoptosis, and blood count were performed only on group II animals from samples that were obtained at moments −8 d, 0 d, +1 d, and +5 d. The 0 d samples were taken from the horses just as they arrived in their stable right after the race. Animals were chosen at random, given the possibility that the horse would take part in a race. According to the rules of the JCSP, once the animal has been included in a race, it can’t undergo any invasive procedure. Thus, a change was made in the collection protocol of group II with the objective of decreasing the interval between the day of collection referred to as a day of performing exercise of lower intensity (from −11 d to −8 d) and the day of the race (0 d) as well as increasing the interval between the sample collected on the day of the race and those that would represent the days of recovery (from +1 d, +2 d, +3 d to +1 d, +5 d).

### 2.4. Sample Collection

The samples were collected via venipuncture with tubes containing sodium heparin for phagocytosis and oxidative burst assay; tubes with EDTA for apoptosis, lymphoproliferation and blood count assays; and the tube without anticoagulant was obtained for serum cortisol assay. Collection times were patronized in order to respect the variation of blood cortisol concentration throughout the day, with higher values in the morning and lower values in the late afternoon, with samples from group 1 animals taken in the moments −11 d, −8 d, 0 d, +1 d, +2 d, +3 d, and +5 d, while sampling for group 2 collection was performed at the moments −8, 0 d, +1 d, and +5 d.

### 2.5. Flow Cytometry Assays

Quantification of phagocytosis and oxidative burst was estimated using mean PI and DCFH fluorescence cell. Briefly, 100 mL whole blood (2 × 10^5^ cells/100 μL) was mixed with 200 mL of DCFH-DA (0.3 mM) in PBS and 100 mL PI-labeled *S. aureus* in polypropylene tubes. Samples were incubated under agitation at 37 °C for 20 min. Reactions were stopped by adding 2 mL of cold EDTA solution (3 mM) to stop phagocytosis. After centrifugation (250× *g* for 10 min), erythrocytes were lysed from all the samples with a red blood lysis solution. Samples were then centrifuged (e.g., 250× *g*, in 0,5 mL concentration of 2 × 10^5^ cells) and the cell pellets resuspended in 0,5 mL of cold EDTA (3 mM) for flow cytometry. Direct measurements of mean fluorescence of green and red channels were recorded as oxidative burst and phagocytosis, respectively, as proposed by [21]. The percentage of phagocytosis (percentage of neutrophils or monocytes which ingested bacteria) is expressed as the number of neutrophils or monocytes with red fluorescence divided by the total number of cells (multiplied by 100).

For the apoptosis assay—detection of apoptosis phenomenon and differentiation of apoptotic cells from necrotic cells—the technique used was the association of an early apoptosis stage indicator, fluorescein isothiocyanate-labeled annexin V (FITC), with a measure of plasma membrane integrity, propidium iodide (PI), allowing for discrimination between living and dead cells. For the lymphoproliferation assay, the separation of circulating mononuclear cells from the peripheral blood was performed by means of density gradient. The adjustment of the cell concentration for subsequent in vitro culture was adapted from CHEN et al. (2003) in 1 × 10^7^ cells/mL of lymphocytes labeled with carboxyfluorescein succinimidyl ester (CFSE-final concentration of 5 μm/mL in PBS) for 20 min at 37 °C and protected from light. After labeling, cells were incubated for 5 min in the dark at 20 °C, and then, 10 mL of RPMI with 5% FBS (fetal bovine serum, Sigma-Aldrich, Missouri, USA) was added and centrifuged at 300× *g* for 5 min at 20 °C. Cells were washed three times and 1 mL RPMI per 5% FBS was added. The cells were plated in a 96-well U bottom plate at a concentration of 2 × 10^5^ cells/well in triplicate with either, medium alone, as well as stimulation with Concanavalin A (ConA, Sigma, Kawasaki, Japan, 5 µg/mL), at 5% CO_2,_ at 37 °C, for 72 h at 37 °C in a 5% CO_2_ atmosphere. At the end of this period, the cells were collected, and the acquisition of the events was performed on the FACSCalibur flow cytometer (Becton & Dickinson, Franklin Lakes, NJ, USA). For each sample, at least 30,000 signals were analyzed and during data acquisition a gate was drawn to select lymphocytes that were identified based on cell characteristic properties in the forward (FSC) and side (SSC) scatter, and a histogram of green fluorescence (FL-1 channel) was defined within the lymphocyte cell gate. The analysis of ConA-stimulated cells within gated cells was used to calculate the percentage of T cells that had moved from the resting to the blast population and the proliferation index of mitotic cells was calculated using Flow Jo cell cycle analysis software (BD Biosciences, Franklin Lakes, NJ, USA). As for the determination of the serum cortisol concentration, this was determined by means of competition radioimmunoassay (commercial kit Coat-A-Count/DPC MEDLAB, Los Angeles, CA, USA). The cortisol detection limit was 0.04 µg/dL, the intra-assay variation coefficients were below 10%, and the inter-assay variation coefficients were between 0.33% and 1.42%.

### 2.6. Statistical Analysis

It was verified, by means of the Bartlett test, which type of parametric or non-parametric test would be the most suitable for each experimental situation; from there, the analysis of variance (ANOVA) followed multiple test comparisons of Tukey–Kramer for the detection of possible differences between the results found in the different collection moments, using JMP software, version 5.0.1.2 (SAS Institute, Cary, NC, USA, 2003).

## 3. Results

For the phagocytosis and oxidative burst assays, cells from the group located in gate 1 (R1) were used, referring to the neutrophil population, while cells located in the region of gate 2 (R2), that concentrated lymphocytes, were used for the apoptosis assay. In gate 3 (R3), the monocyte population was located (Figure 1). The analysis of the results of the phagocytosis, oxidative burst and serum cortisol assays, the moment immediately after the race and a day after the race (common to the two groups) showed statistically significant difference, between groups with *p* < 0.0001; making it impossible to analyze the data together.

Table 2 shows the mean values, standard deviations and the statistical significance considered regarding the phagocytosis of neutrophil assays in the samples taken right after the race (0 d) and one day after it (+1 d). Figure 2A illustrates the difference found. As for the intensity of phagocytosis (number of phagocyte bacterial particles), there was a statistically significant difference between the values obtained at the different moments. The moment immediately after the race and the +3 d moment presented lower values than the moment −11 d and the moment +1 d (Table 2 and Figure 2C). In animals 16 to 30, a decrease was also found in the percentage values of neutrophil phagocytosis immediately after the race, when compared to the other studied moments, −8 d, +1 d, and +5 d, showing a statistical significance (Table 3 and Figure 2B). The average values of phagocytosis intensity obtained from samples collected immediately after the race were lower than those found at +1 d and +5 d, which in turn were lower than the values found at −8 d (Table 3 and Figure 2D).

The mean values found in the oxidative burst assay for the neutrophils from the samples obtained from animals 1 to 15 are shown in Table 4 and Figure 2E. There was a decrease in the values of the samples collected immediately after the race and at the +3 d moment, in relation to the moments referred to as −11 d and +2 d. Regarding the values obtained from animals 16 to 30, no significant statistical differences were found (Table 5 and Figure 2).

For the apoptosis analysis, in the apoptosis assay, the cells located in the region demarcated by gate 1 (R1) were used (Figure 1). Table 6 shows the means and standard deviations for the apoptosis test performed on samples from Group II animals. No significant statistical difference was demonstrated between apoptotic cell values at the different moments of collection (Figure 3). There was a significant difference in relation to the values of necrotic cells; since higher values were found at −8 d, when compared to the moment immediately after the race. The values were lower also in the +1 d and +5 d moments, compared to the moment immediately after the race. As for viable cells, higher values were found at time +5 d and lower values were found at time −8 d when compared to the moment immediately after the race and at time +1 d (Table 6).

Regarding the lymphoproliferation assay, Figure 4 shows the gate used to select the lymphocyte population within the mononuclear population placed in culture, to be analyzed after the test.

The kinetics of the lymphocyte population proliferation assay were analyzed using histograms. Figure 5 represents the kinetics of the control tube (without the use of mitogens with stimulus). The fluorescence presented by the cells was homogeneously distributed, indicating absence of cell division. Markers (M) above the peaks were used to quantify the events in each cell division cycle. The representation of the kinetics of the lymphoproliferation assay after the use of Concanavalin A (ConA) as a stimulus is shown in Figure 6. The data were used to construct Table 7. No significant statistical difference was found when comparing the values found at the different moments of sample collection.

The blood count data found at the sample collection moments for group II were collected in Table 8. The percentage of neutrophils was increased immediately after the race, when compared to the other moments (Figure 7). The percentage number of lymphocytes at the time +5 d was increased, compared to the other moments (Figure 8). The total number of erythrocytes (He) for different collection days, when compared to each other, showed a significant difference in erythrogram values; as well as hemoglobin concentration (Hb), globular volume (Hct), mean corpuscular volume (MCV), mean corpuscular hemoglobin (MCH), and mean corpuscular hemoglobin concentration (MCHC).

Table 9 represents mean values and the standard deviation of serum cortisol levels present in serum of group I animals. The mean values were increased immediately after the race, in comparison to the values of the other moments (Figure 9). Table 10 shows the values of serum cortisol concentration obtained in the samples of animals in group II, at different collection moments. The increased mean values were also found in the samples collected immediately after the race (Figure 10).

## 4. Discussion

In the present work neutrophils phagocytosis assay, oxidative burst, lymphocyte proliferation, red blood count, blood cortisol evaluation, and apoptosis assays were performed in thirty Thoroughbred athlete horses after training, racing and during resting periods. The horses were divided in two groups with samples taken at different times. To validate the investigation a preliminary analysis was performed comparing the results obtained from the samples taken immediately after the race (moment 0 d) and a significant statistical difference was found between the two groups (*p* ˂ 0.001). This phenomenon could be correlated to the fact that group II was a more cohesive group regarding race category, since most of the animals were participants of Grand Prix, indicating that they presented at least one victory in their history and consequently could perform better.

The studied horses raced distances varying between 1400 m and 2400 m, in this enabling a consumption of energy from both anaerobic and aerobic supplies [9,22]. The average running time for the 1400 m was 1′25″403, while the 2400 m runs averaged 2′29″595. In order to accommodate the large increase in the demand of energy and oxygen occurring during a high-intensity exercise for muscle work and to allow the dissipation of metabolism and heat products, the processes are initiated and controlled by neural factors, hormones, hypoxia condition, and vasoactive peptides released by muscles [23]. During such intense exercises the most common findings are an increase in O_2_ transportation correlated with increased blood cell parameters and gene expression [8].

The improvement in the oxygen release when facing intense muscle work is started by the activation of the sympathetic nervous system and the hypothalamus-pituitary-adrenal axis (HPA). There is a quick increase in the circulation of adrenocorticotropic hormone (ACTH), adrenaline, noradrenaline, and cortisol [23]. When the HPA axis is activated during exercise, glucocorticoid hormones act as the final effector molecules, promoting, when released, an increase in liver glycogenesis and induced lipolysis, providing fuel for prolonged submaximal exercises. However, the hormones act through negative feedback on the immune system, suppressing its functions [24]. However, it is important to clarify that data regarding immune changes and blood parameter differ greatly in the available literature [10].

The hormonal dosage of serum cortisol levels displayed an increase in the samples obtained from the two groups of horses collected soon after the race when compared to the results of the samples collected at other times (Table 9 and Table 10 and Figure 9 and Figure 10). This result also coincides with a decrease in neutrophil function, suggesting that the greatest stress moment for the animals was during the race period (Table 2 and Table 3 and Figure 2A–F). Cappeli et al. [8] already correlated the increase in cortisol secretion with training of young Thoroughbreds, in this way the results found match what was proposed in their work.

An increase in the relative number of neutrophils was observed in the samples collected immediately after the race using the blood count (Table 8 and Figure 7). It is known that this initial increase stems from the mobilization of marginalized segmented neutrophils, induced by the release of catecholamines, while a late increase results from the mobilization of bone marrow neutrophils for circulation, as the result of a release induced by increased plasma cortisol levels [8,13]. As for the neutrophil and lymphocyte rate, the results showed no significant statistical difference when compared to the other moments.

Nonetheless, there was a slight increase in the rate of these cells soon after the race that could reflect the group II animals’ conditioning status, that could only be confirmed with the comparison of all collection moments with a group known to be less conditioned. However, the work of Cywinska et al. [7] verified that neutrophil activity was higher after training in racehorses, suggesting the conclusion made by the researchers that these activities didn’t result in impaired immune functions for the studied animals [7].

Exercise, therefore, seems to result in an initial activation of neutrophils, with a variable response in effector functions, such as phagocytic activity and oxidative burst capacity, depending on the intensity of the physical activity practiced, as already suggested [2,25]. Twenty-four hours after the competition, the phagocytic activity of the neutrophils showed values close to those found in the samples collected from eleven (−11 d) to eight (−8 d) days before the event, both in horses belonging to group I and II. Regarding the oxidative burst activity, its evaluation could only be performed in group I. These data corroborate with the results of a study carried out in humans in which the oxidative capacity of athletes’ neutrophils was compared to moderate and high intensity activity exercises [18].

An initial decrease in the activity of neutrophils was found in both situations that remained during the high intensity exercise. Meanwhile, in the moderate intensity exercise and after a few hours of the activity, there was an increase in the functional capacities of these cells. The cause of phagocytic suppression is multifactorial but can be partially attributed to an increase of plasma cortisol immediately after racing. Further, the decrease in neutrophil function may reflect a higher percentage of immature neutrophil migration from the bone marrow to circulation [13,26]. Moreover, some studies have already associated changes in the innate function, respiratory infections, and conditioning status of horses [20,27].

In relation to lymphocytes, no significant statistical difference was found when comparing the values analyzed from the different collection moments. However, since the trial was only conducted with samples collected from group II animals (which was randomly comprised only of grand prize race participants), this fact could be attributed to the training status and efficiency of these animals, as suggested by some authors, or to a better adaptation to stress [28,29]. As for the number of apoptotic cells, in this study, there was no statistical difference between the values found, a finding that indicates a lack of major damage to the cells; suggesting that the animals achieved good conditioning to stress. In addition, it is possible to relate the phenomenon with the fact that high-intensity activities can increase concentrations of anti-inflammatory cytokines, which aims to decrease the damage to muscle tissue resulting from its inflammation, also promoting a higher susceptibility to infections [30]. Metabolic and hormonal decompensation can induce apoptosis in in vitro experiments, as well as changes in cytosolic calcium concentration that also occurs during exercise and can also represent an apoptosis deflagration pathway [31]. The modifications that occur in the oxide-reduction state may be another important path with evidence that the apoptosis process can be induced by an alternative pathway during exercise, such as changes in the expression of the CD95 receptor (Fas/APO-1) [31,32].

Regarding the erythrocytic series, the increased values found match the data found in the literature, that also reports higher clotting parameters for athlete Thoroughbreds [10]. Given the requirement of muscular work, the equine body releases into circulation erythrocytes stored in the spleen, in substantial quantities, in order to attend the need of higher transportation of O2 [8]. The importance of these actions is related to the concentration of blood oxygen increase, in about 100 mL/L responding to maximum exercise, also associated with a rise in hematocrit between rest and maximum [33]. In the case of conditioned horses, some effects of training should be considered, such as anabolic adaptations like the protein mass increases as a result of constant exercise, and variations in enzymatic pattern. The mechanisms involved in these training effects are not well understood yet. However, there seems to be an involvement of neuroendocrine mechanisms both in the response to acute exercises and adaptations to chronic training [23]. Some studies have demonstrated some resulting adaptations that can influence immune function [34]. Anyhow, the comparison is impaired for several reasons, such as training protocol variance in types and intensity of exercises used, interference of genetic factors, and environmental conditions.

In the horses studied, although a transient decrease in neutrophil function was found, in accordance with recent study reports, the adaptive function was not affected, a finding that could be related to the high intensity of the race, preventing greater damage to the better conditioned animals from group II [35]. Given the neutrophil transient reduction of immune function, it is reasonable that the veterinarians responsible for racing horses perform a more rigorous clinical follow-up of the animals in the days following the races and to monitor the serum concentration of acute phase proteins, particularly serum amyloid A (SAA) [36,37]. SAA is a major acute phase protein, being correlated with high intensity training in inexperienced horses that can suffer microinjuries and muscular glycogen depletion in the beginning of their career and adaptation to exercise [9].

These findings suggest that training may exert a conditioning on gene expression at rest, leading to a faster response to exercise-induced stress in Thoroughbreds, proposing that a constant and focused training regime could increase the baseline expression of genes involved in the inflammatory process. The most analyzed genes in research about stress and exercises in horses are *IL-4*, *IL-6*, *INF-**γ*, and *TNF-**α*, with their high expression after exercise indicating metabolic changes and immunomodulation [2,8,9]. This could prepare horses for acute stress and limit the subsequent inflammatory reaction, thereby permitting a better response to exercise-induced stress and a reduced likelihood of low-performance syndromes [38]. The analysis of molecular mechanisms from exercise-induced adaptations that are important to running performance may culminate in new evidence-based training methodologies, therapeutic strategies for diseases, lower rates of catastrophic injuries, and improvement in animal welfare [39].

## 5. Conclusions

Given the increase in the percentage of neutrophils in the blood count and the influence of serum cortisol elevation, it was possible to correlate the integration of the neuroendocrine and immune systems in response to the high intensity exercises performed during races. The transient decrease found in the in vitro function of neutrophils through phagocytosis and oxidative burst assays reveals the activation of innate immune cells in the face of exercise-induced stress. The absence of alterations in lymphocyte proliferation values before antigenic stimuli in vitro suggests that adaptive immunity has not been affected. Likewise, the absence of lymphocyte found in the process of apoptosis can show that there has been no major damage to specific immunity cells. The values found in the blood count, in relation to the erythrocytic series, may demonstrate the adaptation of the horses to the demands of the work performed during the exercise.

## Figures and Tables

**Figure 1 animals-12-01203-f001:**
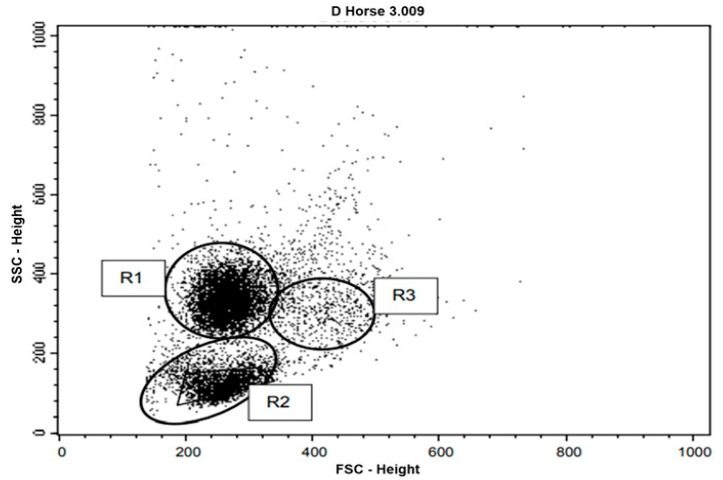
Representation of peripheral blood leukocyte flow cytometry data of the studied horses, after lysis of erythrocytes, in a dot plot. Cell populations were selected according to their forward scatter (FSC) versus side scatter (SSC) profiles indicating cell size versus granularity and complexity. R1 corresponds to the neutrophil gate, R2 corresponds to lymphocytes, and R3 corresponds to monocytes.

**Figure 2 animals-12-01203-f002:**
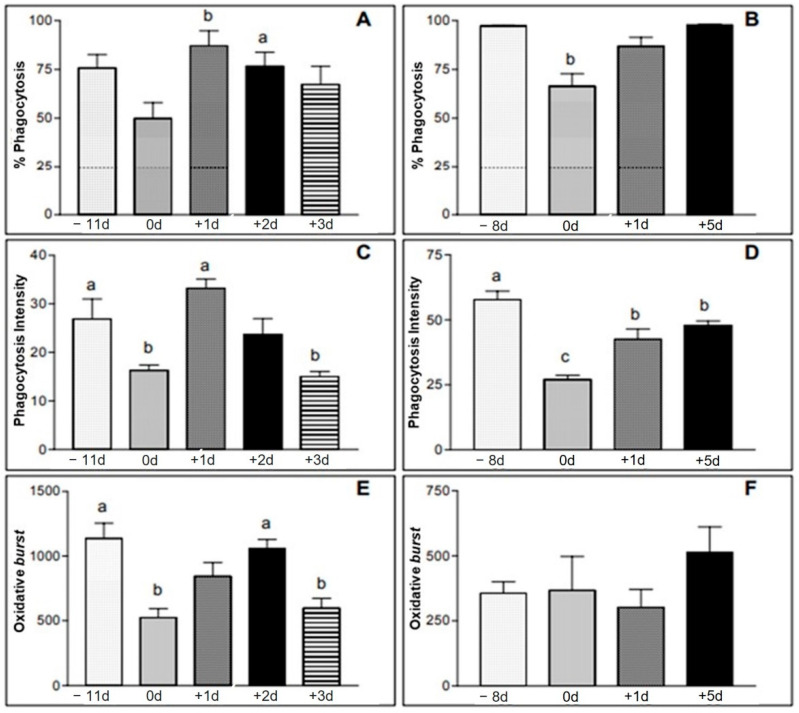
(**A**,**B**): % of phagocytosis performed by neutrophils from animals of group I (**A**) and group II (**B**) samples. (**C**,**D**): phagocytosis intensity expressed as mean fluorescence intensity (geometric mean of the number of bacteria destroyed by neutrophils) from animals in group I (**C**) and group II (**D**). (**E**,**F**): mean fluorescence on a logarithmic scale of the neutrophils in the samples of animals of group I (**E**) and group II (**F**). Group I—moments −11 d, 0 d, +1 d, +2 d and +3 d. Group II—moments −8 d, 0 d, +1 d and +5 d. a,b = significant statics difference.

**Figure 3 animals-12-01203-f003:**
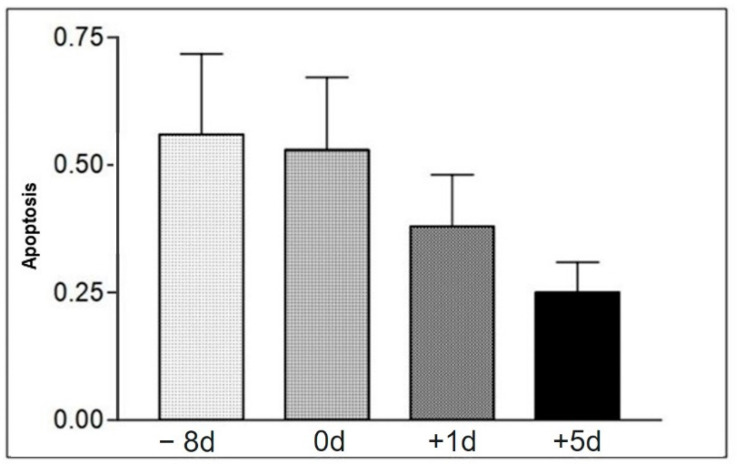
Mean values of total apoptotic cells in moments −8 d, 0 d, +1 d, and +5 d, in the samples of the animals of group II. Statistical difference according to the Tukey–Kramer test—Beltsville, 2003.

**Figure 4 animals-12-01203-f004:**
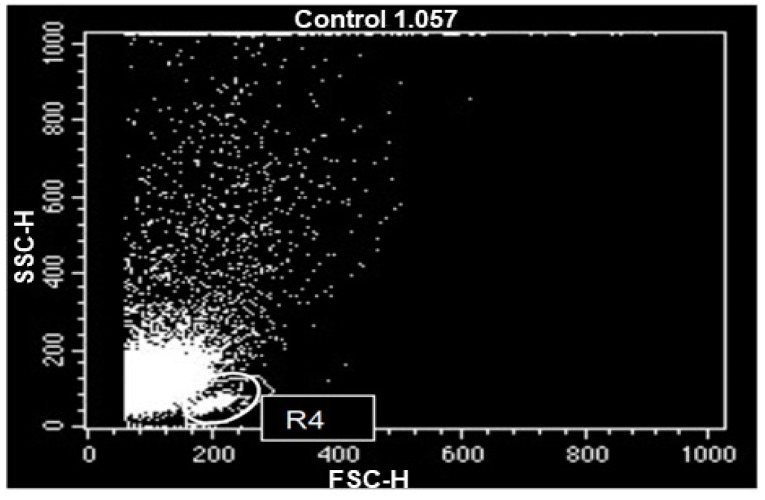
Flow cytometry data, in dot plot, of the peripheral blood mononuclear leukocytes of the studied horses, after a 72-h culture. Cell populations identified according to their forward scatter (FSC) versus side scatter (SSC) profiles, cell size versus granularity and complexity. R4—lymphocyte gate.

**Figure 5 animals-12-01203-f005:**
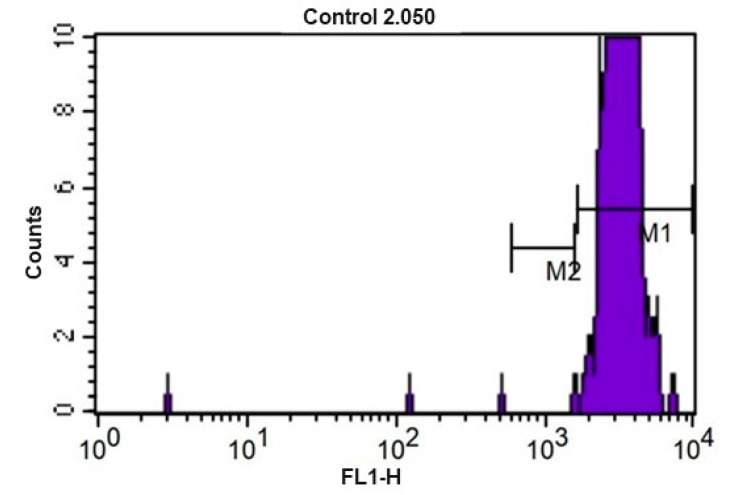
CFSE fluorescence histogram (carboxyfluorescein diacetate) of equine peripheral blood lymphocytes after culture for 72 h at 5% CO_2_ and 37 °C without use of mitogens (control). M1 and M2 represent the markers used to enumerate the events of each division cycle that the software used for calculating the proliferation index.

**Figure 6 animals-12-01203-f006:**
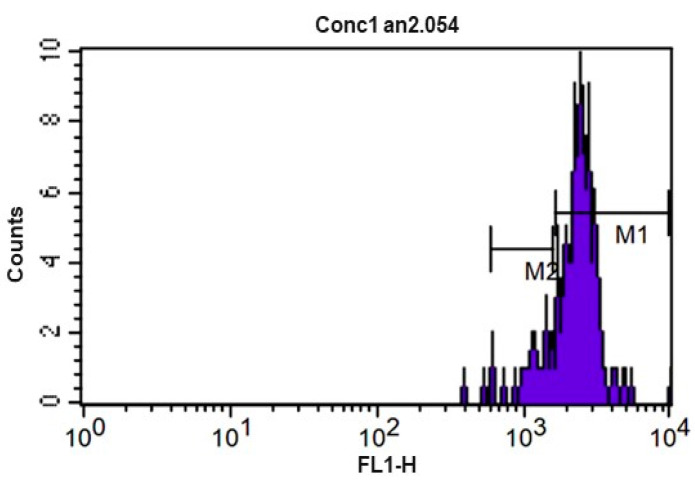
CFSE fluorescence histogram (carboxyfluorescein diacetate) of lymphocytes from equine peripheral blood after 72 h culture at 5% CO_2_ and 37 °C without concavalin A. M1 and M2 represent the markers used to list the events of each division cycle that the software used for calculating the proliferation index.

**Figure 7 animals-12-01203-f007:**
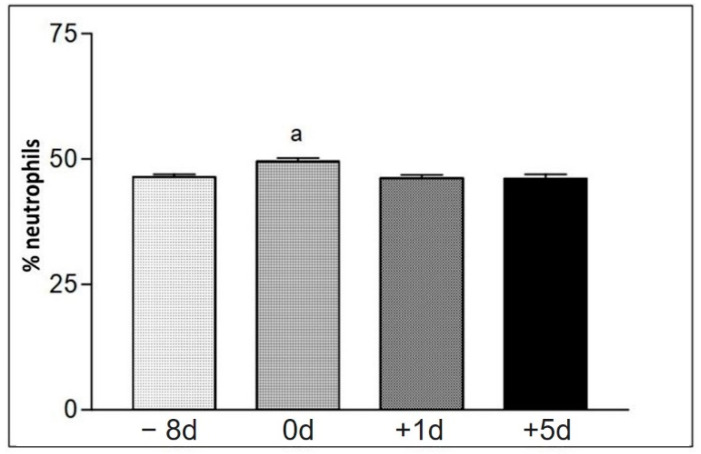
Representation of the average values referring to the counting of the relative number (%) of neutrophils (No) at the different collection days defined as −8 d, 0 d, +1 d, and +5 d. Statistical difference according to the Tukey–Kramer test—Beltsville, 2003. a = significant statics difference.

**Figure 8 animals-12-01203-f008:**
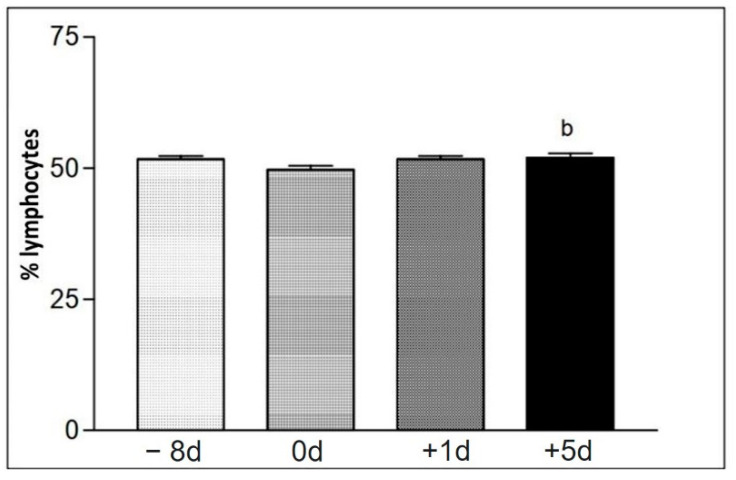
Representation of the average values of the relative number (%) of lymphocytes (Lo) on the different collection days defined as −8 d, 0 d, +1 d, and +5 d. Statistical difference performed according to the Tukey–Kramer test—Beltsville, 2003. b = significant statics difference.

**Figure 9 animals-12-01203-f009:**
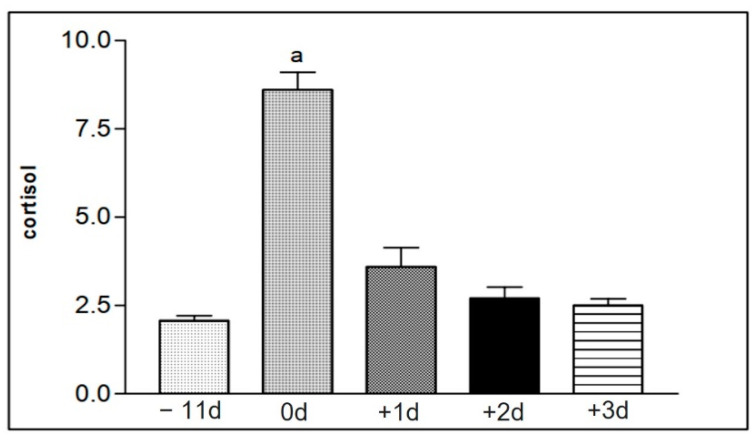
Representation of the mean values of serum cortisol levels from the samples of the animals in group I (1 to 15), taken at the different collection moments at −11 d, 0 d, +1 d, +2 d, and +3 d. Statistical difference according to the Tukey–Kramer test—Beltsville, 2003. a = significant statics difference.

**Figure 10 animals-12-01203-f010:**
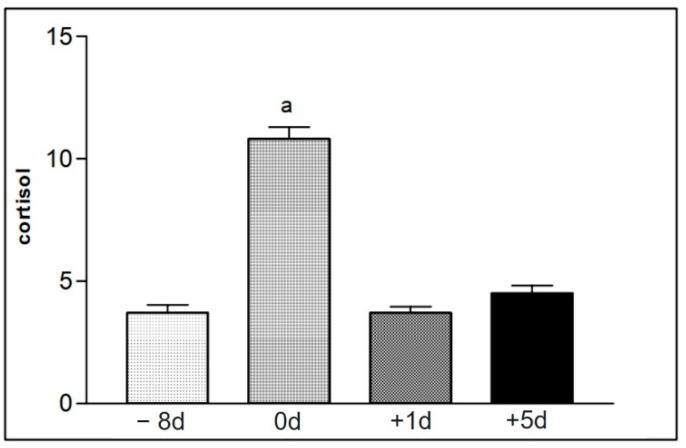
Representation of the mean values of serum cortisol levels of samples from animals in group II (16 to 30), found at the different moments of collection defined as −8 d, 0 d, +1 d, and +5 d. Statistical difference according to the Tukey–Kramer test—Beltsville, 2003. a = significant statics difference.

**Table 1 animals-12-01203-t001:** Studied animals’ sex, age (in years), and distance raced in meters.

Animal	Sex	Age	Distance Raced
1	F	2	1400 m
2	M	4	1600 m
3	F	3	1400 m
4	F	3	1800 m
5	M	4	1600 m
6	M	3	2400 m
7	F	3	1800 m
8	F	3	1800 m
9	M	4	1500 m
10	M	5	1600 m
11	M	2	1500 m
12	M	3	1400 m
13	F	5	2000 m
14	F	3	1500 m
15	F	3	1500 m
16	M	5	1400 m
17	F	3	1400 m
18	F	3	1400 m
19	M	3	1400 m
20	M	3	1600 m
21	M	3	1600 m
22	M	5	2400 m
23	M	4	2400 m
24	M	6	2400 m
25	F	3	2000 m
26	F	3	2000 m
27	M	4	2400 m
28	M	3	2000 m
29	M	3	2000 m
30	M	4	2400 m

**Table 2 animals-12-01203-t002:** Values representing the mean and the standard deviation of the percentage of cells that underwent phagocytosis and its intensity, with blood samples from group I (1 to 15) at times −11 d, 0 d, +1 d, +2 d, and +3 d. Statistical difference according to the Tukey–Kramer test—Beltsville, 2003.

Group I	−11 d	0 d	+1 d	+2 d	+3 d
% Phagocytosis	75.8 ± 26.0 ^ab^	49.9 ± 27.8 ^b^	87.1 ± 18.9 ^a^	76.7 ± 24.9 ^ab^	67.3 ± 24.8 ^ab^
Intensity	26.9 ± 15.4 ^a^	16.3 ± 4.3 ^b^	33.2 ± 3.9 ^a^	23.7 ± 11.8 ^ab^	15.0 ± 3.3 ^b^

Distinct letters on the same line indicate a significant statistical difference between the results, 0 d = immediately after the race.

**Table 3 animals-12-01203-t003:** Values representing the average and the standard deviation of the percentage of cells that underwent phagocytosis and their intensity, analyzed with blood samples taken from animals in group II (16 to 30) at moments −8 d, 0 d, +1 d, and +5 d. Statistical difference according to the Tukey–Kramer test—Beltsville, 2003.

Group II	−8 d	0 d	+1 d	+5 d	Significance
% Phagocytosis	97.4 ± 1.2 ^a^	66.4 ± 19.5 ^b^	86.9 ± 13.6 ^a^	97.8 ± 1.1 ^a^	*p* < 0.0001
Intensity	57.8 ± 9.1 ^a^	27.0 ± 4.8 ^c^	42.6 ± 11.7 ^b^	47.9 ± 4.4 ^b^	*p* < 0.0001

Distinct letters on the same line indicate a significant statistical difference between the results, 0 d = immediately after the race.

**Table 4 animals-12-01203-t004:** Values representing the average and the standard deviation of the oxidative burst performed with blood samples collected from animals in group I (1 to 15) at the moments −11 d, 0 d, +1 d, +2 d, and +3 d. Statistical difference measured according to the Tukey–Kramer test—Beltsville, 2003.

Group I	−11 d	0 d	+1 d	+2 d	+3 d	Significance
Oxidative burst	1136.4 ± 407.2 ^a^	524.2 ± 248.9 ^b^	844.1 ± 256.6 ^ab^	1057.6 ± 253.1 ^a^	596.3 ± 231.6 ^b^	*p* < 0.0001

Distinct letters on the same line indicate a significant statistics difference between the results, 0 d = immediately after the race.

**Table 5 animals-12-01203-t005:** Values representing the average and the standard deviation of oxidative burst performed with blood samples collected from animals in group II (16 to 30) at the moments −8 d, 0 d, +1 d, and +5 d. Statistical difference according to the Tukey–Kramer test—Beltsville, 2003.

Group II	−8 d	0 d	+1 d	+5 d	Significance
Oxidative burst	356.6 ± 123.1 ^a^	367.4 ± 345.2 ^a^	302.0 ± 169.0 ^a^	514.2 ± 217.7 ^a^	*p* > 0.4

Distinct letters on the same line indicate a significant statistic difference between the results, 0 d = immediately after the race.

**Table 6 animals-12-01203-t006:** Values representing the average and the standard deviation of the results of the apoptosis assay performed with blood samples collected from animals from group II (16 to 30), at moments −8 d, 0 d, +1 d and +5 d. Statistical difference according to the Tukey–Kramer test—Beltsville, 2003.

Group II	−8 d	0 d	+1 d	+5 d	Significance
Apoptosis	0.56 ± 0.61 ^a^	0.53 ± 0.55 ^a^	0.38 ± 0.39 ^a^	0.25 ± 0.23 ^a^	*p* > 0.5
Necrosis	0.89 ± 0.28 ^a^	0.57 ± 0.58 ^ab^	0.31 ± 0.17 ^b^	0.23 ± 0.15 ^b^	*p* < 0.001
Viable cells	98.6 ± 0.6 ^b^	98.7 ± 0.7 ^ab^	99.3 ± 0.5 ^ab^	99.4 ± 0.4 ^ab^	*p* < 0.009

Distinct letters on the same line indicate a significant statistic difference between the results, 0 d = immediately after the race.

**Table 7 animals-12-01203-t007:** The values represent the average and standard deviation of lymphoproliferation results performed with blood samples collected from animals in group II (16 to 30) at −8 d, 0 d, +1 d, and +5 d. Statistical difference according to the Tukey–Kramer test—Beltsville, 2003.

Group II	−8 d	0 d	+1 d	+5 d	Significance
Lymphoproliferation	30.9 ± 4.0 ^a^	28.4 ± 4.5 ^a^	29.8 ± 3.3 ^a^	30.8 ± 3.6 ^a^	*p* > 0.2

Distinct letters on the same line indicate a significant statistical difference between the results, 0 d = immediately after the race.

**Table 8 animals-12-01203-t008:** Values represent the average and standard deviation of the blood count results of the horses analyzed in the study: counting the total number of leukocytes (Leu); the relative number (%) of neutrophils (No), eosinophils (Eo), monocytes (Mo), and basophils (Bo); the total number of erythrocytes (He); hemoglobin concentration (Hb); globular volume (Hct); mean corpuscular volume (MCV); mean corpuscular hemoglobin (MCH); and mean corpuscular hemoglobin concentration (MCHC) at different collection times defined as −8 d, 0 d, +1 d, and +5 d. Statistical difference according to the Tukey–Kramer test—Beltsville, 2003.

Group II Blood Count	−8 d	0 d	+1 d	+5 d	Significance
Leu (10³/mL)	7927 ± 1389 ^a^	9547 ± 1640 ^a^	8793 ± 1677 ^a^	8473 ± 1677 ^a^	*p* > 0.09
No (%)	4.6 ± 2.3 ^b^	49.5 ± 2.9 ^a^	46.2 ± 2.4 ^b^	46.1 ± 3.3 ^b^	*p* < 0.003
Lo (%)	51.7 ± 2.4 ^a^	49.7 ± 2.9 ^a^	51.7 ± 2.5 ^a^	52.0 ± 3.2 ^b^	*p* > 0.09
Mo (%)	0.0 ^a^	0.0 ^a^	1.0 ^a^	0.0 ^a^	*p* > 0.2
Eo (%)	1.33 ± 0.48 ^a^	1.06 ± 0.26 ^a^	1.33 ± 0.19 ^a^	1.53 ± 0.52 ^a^	*p* > 0.06
Bo (%)	0.0 ^a^	0.0 ^a^	0.0 ^a^	0.0 ^a^	*p* > 0.2
He (×10³)	10,378 ± 643.6 ^b^	13,021 ± 1066 ^a^	11,231.3 ± 993.3 ^b^	10,658.6 ± 886.2 ^b^	*p* < 0.0001
Hb (g/dL)	14.6 ± 0.8 ^b^	17.5 ± 1.3 ^a^	15.3 ± 0.9 ^b^	14.8 ± 0.9 ^b^	*p* < 0.0001
Hct (%)	43.5 ± 3.1 ^b^	57.7 ± 6.7 ^a^	46.7 ± 3.8 ^b^	44.6 ± 3.8 ^b^	*p* < 0.0001
MCV	41.9 ± 0.7 ^b^	44.2 ± 2.3 ^a^	41.6 ± 0.7 ^b^	41.8 ± 0.6 ^b^	*p* < 0.0001
MCH	13.9 ± 0.2 ^a^	13.5 ± 0.3 ^b^	12.7 ± 0.4 ^ab^	13.9 ± 0.3 ^a^	*p* < 0.001
MCHC	33.3 ± 0.7 ^a^	30.5 ± 1.6 ^b^	32.8 ± 0.6 ^a^	33.3 ± 0.8 ^a^	*p* < 0.0001

Distinct letters on the same line indicate a significant statistical difference between the results, 0 d = immediately after the race.

**Table 9 animals-12-01203-t009:** The values represent the average and standard deviation from serum cortisol concentrations, assessed by the blood samples collected from animals of group I (1 to 15) at the moments −11 d, 0 d, +1 d, +2 d, and +3 d. Statistical difference measured according to the Tukey–Kramer test—Beltsville, 2003.

Group I	−11 d	0 d	+1 d	+2 d	+3 d	Significance
Cortisol	2.07 ± 0.5 ^b^	8.6 ± 1.9 ^a^	3.6 ± 1.2 ^b^	2.7 ± 1.1 ^b^	2.5 ± 0.6 ^b^	*p* < 0.0001

Distinct letters on the same line indicate a significant statistical difference between the results, 0 d = immediately after the race.

**Table 10 animals-12-01203-t010:** The values represent the average and standard deviation of the concentrations of serum cortisol measured with blood samples collected from animals in group II (16 to 30) at the moments −8 d, 0 d, +1 d, and +5 d. Statistical difference according to the Tukey–Kramer test—Beltsville, 2003.

Group II	−8 d	0 d	+1 d	+5 d	Significance
Cortisol	3.7 ± 1.3 ^b^	10.8 ± 1.9 ^a^	3.7 ± 1.0 ^b^	4.5 ± 1.2 ^b^	*p* > 0.0001

Distinct letters on the same line indicate the significant statistics difference between the results, 0 d = immediately after the race.

## Data Availability

All important data were added to the manuscript.

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
