# Peer review of "Immune Functions Alterations Due to Racing Stress in Thoroughbred Horses"

_animals, 2022, doi:10.3390/ani12091203_

Round 1

Reviewer 1 Report

Review of the article "Immune functions alterations due to racing stress in Thouroughbred horses".

General comments

The manuscript is clear, relevant for the field and presented in a well-structured manner. The experimental design is appropriate to test the hypothesis.

Specific comments

Some figures could be deletes

In the discussion lines 501-514 could be deleted.

Author Response

We would like to thank you for the considerations and advices. The response to all questions and comments was added below.

  • General comments

The manuscript is clear, relevant for the field and presented in a well-structured manner. The experimental design is appropriate to test the hypothesis.

R: Thank you for the insights!

Specific comments

1- Some figures could be deletes

R: The kept figures were better discussed in the results.

2- In the discussion lines 501-514 could be deleted

R: In accordance, the referred lines were deleted.

Reviewer 2 Report

GENERAL COMMENTS

Conducting field studies is always difficult and the authors are to be congratulated for this attempt. However, I had some difficulty in identifying an explicit hypothesis to be tested. Depending on the section considered, the objective of the study is expressed differently.

Abstract

“For the present investigation alterations of the number and function of cellular populations responsible for the homeostasis re-establishment, through immune defense mobilizing neutrophils and lymphocytes, were verified”

Introduction

“Due to this it is important to monitor the immune system disruptions during training and racing, always intending to preserve the animal wellbeing and the system homeostasis. Addressing this issue, ...”

Discussion

This study aimed to verify the influence of environmental factors associated with high performances competitions, as well as factors associated only with high intensity exercises competitions days, on the in vivo immune function and blood components of Thoroughbreds.”

Were “environmental factors” evaluated?

There are several little misprints in the manuscript (e.g. 250 x g, in 0,5 ml, concentration of 2x105 cells, etc.) to be corrected. To separate decimals, commas should be replaced by full stops throughout the text and in tables. Formatting of the references must be adapted to that requested by the guidelines of ‘Animals’ and must be standardised. Some references mentioned in the text are lacking (e.g. Pyne, 1994) .

COMMENTS BY SECTIONS

Introduction:

General comments

The introduction and objectives should be rewritten to better justify the study after stating the current knowledge and knowledge gaps (see general comments above). Several references do not seem the most appropriate to serve the purpose (e.g. Pilaszewicz et al. 2020; Nipin et al. 2020); references to pioneering studies should be preferred to more recent ones that may be less relevant to the subject.

Specific comments

Line 42 to 44: The link between the popularity of racing and the fact that it is an intensive sport is not obvious. Popularity can arise from other aspects (e.g. gambling, associated betting, etc.; see publications on the subject) than ‘intensity’.

Line 48: Robson refers to endurance racing. Please find a reference to Thoroughbred racehorses or specify the discipline.

Line 73: in vitro should be italicized.

Materials and methods:

General comments

The location is of poor interest but delay between sampling and analysis is warrant. This timing may artificially modify the phenotype and functional characteristics of the neutrophils.

Specific comments

Line 114: authors mention “Collection times were patronized, in order to respect the variation of blood cortisol concentration …” but did not specify the time of collection. Was it standardised?

Line 120: Flow cytometry assays – PI is not defined when first met (line 120): do you mean phagocytosis index or as later defined (line 138), propidium iodide (PI) or proliferation index (PI; line 161)? Please, clarify. DCFH is not defined.

Experimental design - The experimental design should be described before the analyses. However, the timing of post-race sampling makes the comparison of 3 days versus 5 days post-race difficult. Also, could you specify what means “immediately after the race (moment 0d)”? Were the horses sampled on the track, at their arrival or when they have reached back their stable? My question is “how many minutes after the end of the race were the horses sampled?”

Line 178: As it is written, I understand that phagocytosis, oxidative burst and serum cortisol concentration were not measured in group II 8 days before the race which is contradicted by measurements in the tables. Is moment -8d lacking line 178? Were lymphoproliferation, lymphocyte apoptosis and blood count not measured in group I? Could you rephrase to facilitate the understanding?

Why did you make two groups? Because it was not possible to measure everything in all of them or simply because you decided to change the timing? Was this timing changed (also) because of preliminary analysis of group I results? As I understand it, the constitution of two groups is not intended to compare them but it is a redesign of the experiment?

Results

Line 205: ‘showed statistically significant difference…’ with what? Among groups?

Line 214: In the phagocytosis assay, there was a decrease … from group I, from 1 to 15. Sorry, I do not understand the sentence nor its correspondence in the table. Please clarify.

Graphs

Change the spelling of all graph legends to English (e.g. phagocytosis vs. fagocytosis)

Graphic 3.2

Legend: A e B to be changed by A and B

Tables

Significance value should only be mentioned when significant

Table 3. 8 – Units are lacking

Discussion

The authors suggest that the difference between the groups would be explained by their previous performance. We lack information regarding the current status of horses: were the horses at the same level of fitness at the time of sampling? Did the two groups participate in the same type of race? Were the two groups investigated in different seasons? Were their training sites similar (i.e. at the Jockey Club of São Paulo (JCSP))?

Many hypotheses are formulated in the discussion to explain the observed variations; several of them are supported by Pyne 1994, a reference that could not be retrieved from the reference list.

Line 440 to 446: although not necessarily untrue, this hypothesis derived from a 40-year-old book chapter is too unspecific to be discussed here while respiratory function is not assessed in these groups of horses

Line 447: it is difficult to relate respiratory infections or conditioning status to results as both have not been documented

Line 498: we may wonder why you did not measured SAA

Remaining parts: again, they are more theoretical statements than hypotheses suggested by the results

Author Response

General comments

Conducting field studies is always difficult and the authors are to be congratulated for this attempt. However, I had some difficulty in identifying an explicit hypothesis to be tested. Depending on the section considered, the objective of the study is expressed differently.

R: Aiming to clarify the hypothesis, we changed the appointed paragraphs in the abstract, introduction and discussion sections, as attached below.

Abstract

  1. “For the present investigation alterations of the number and function of cellular populations responsible for the homeostasis re-establishment, through immune defense mobilizing neutrophils and lymphocytes, were verified”

R: The present paragraph was changed to: The cited paragraph was change for: Aiming to verify the state of blood components and cortisol alterations during the routine of the horses and during races several tests were performed.

Introduction

  1. “Due to this it is important to monitor the immune system disruptions during training and racing, always intending to preserve the animal wellbeing and the system homeostasis. Addressing this issue, ...”

R: The present paragraph was changed to: Considering that stress caused by intense exercise could impact in immune function, it is important to monitor the immune system as well as other blood components and cortisol levels in racing horses. Addressing this issue…

Discussion

  1. This study aimed to verify the influence of environmental factors associated with high performances competitions, as well as factors associated only with high intensity exercises competitions days, on the in vivo immune function and blood components of Thoroughbreds.”

Were “environmental factors” evaluated?

R: The present paragraph was changed to: In the present work alterations in immune cells components and cortisol were verified comparing races that comprised high intensity exercise and training moments in Thoroughbreds.

There are several little misprints in the manuscript (e.g. 250 x g, in 0,5 ml, concentration of 2x105 cells, etc.) to be corrected. To separate decimals, commas should be replaced by full stops throughout the text and in tables. Formatting of the references must be adapted to that requested by the guidelines of ‘Animals’ and must be standardised. Some references mentioned in the text are lacking (e.g. Pyne, 1994) .

R: The commas were substituted, and the measure was corrected. All references were adapted in accordance to the journal format.

COMMENTS BY SECTIONS

Introduction:

General comments

The introduction and objectives should be rewritten to better justify the study after stating the current knowledge and knowledge gaps (see general comments above). Several references do not seem the most appropriate to serve the purpose (e.g. Pilaszewicz et al. 2020; Nipin et al. 2020); references to pioneering studies should be preferred to more recent ones that may be less relevant to the subject.

R: The introduction section was rewritten, and new references were included taking in consideration this comment.

Specific comments

Line 42 to 44: The link between the popularity of racing and the fact that it is an intensive sport is not obvious. Popularity can arise from other aspects (e.g. gambling, associated betting, etc.; see publications on the subject) than ‘intensity’.

R: The present paragraph was changed to: Racing is one of the many equestrians sports having a huge popularity with Thoroughbreds being the most used horses, and for this purpose they are subject to intensive training in order to perform well.

Line 48: Robson refers to endurance racing. Please find a reference to Thoroughbred racehorses or specify the discipline.

R: The mention of the endurance racing was added.

Line 73: in vitro should be italicized.

R: Done.

Materials and methods:

General comments

The location is of poor interest but delay between sampling and analysis is warrant. This timing may artificially modify the phenotype and functional characteristics of the neutrophils.

R: The location section was excluded due to its irrelevance to the analyzes, considering that after collection samples were immediately transported to the laboratory for exam performance.

Specific comments

Line 114: authors mention “Collection times were patronized, in order to respect the variation of blood cortisol concentration …” but did not specify the time of collection. Was it standardised?

R: There were multiple collections done per animal, considering that different moments were analyzed; for each moment samples were taken immediately after the race or training as previously explained in the experimental design section.

Line 120: Flow cytometry assays – PI is not defined when first met (line 120): do you mean phagocytosis index or as later defined (line 138), propidium iodide (PI) or proliferation index (PI; line 161)? Please, clarify. DCFH is not defined.

R: PI means Propidium Iodite and the 161 line definition as phagocytosis indexed was excluded. DCFH means fluorescence of dichlorofluorescein. These significates were added in the text.

Experimental design - The experimental design should be described before the analyses. However, the timing of post-race sampling makes the comparison of 3 days versus 5 days post-race difficult. Also, could you specify what means “immediately after the race (moment 0d)”? Were the horses sampled on the track, at their arrival or when they have reached back their stable? My question is “how many minutes after the end of the race were the horses sampled?”

R: The experimental section was moved to item 2.3, right before the sample collection section. The time of collection varied between horses depending on how much after the race they arrived in the stalls, however samples were taken just as they arrived in their paddocks. A paragraph was added to this section explaining this time frame.

Line 178: As it is written, I understand that phagocytosis, oxidative burst and serum cortisol concentration were not measured in group II 8 days before the race which is contradicted by measurements in the tables. Is moment -8d lacking line 178? Were lymphoproliferation, lymphocyte apoptosis and blood count not measured in group I? Could you rephrase to facilitate the understanding?

R: Yes, the -8d was lacking but it was already added in the text. Lymphoproliferation, lymphocyte apoptosis and blood count were performed only in group I

Why did you make two groups? Because it was not possible to measure everything in all of them or simply because you decided to change the timing? Was this timing changed (also) because of preliminary analysis of group I results? As I understand it, the constitution of two groups is not intended to compare them but it is a redesign of the experiment?

R: The groups were divided randomly considering the probability of the horses racing in the subsequently week. The group II samples collection timing was done in this way in order to diminish the period between the low exercise day -11d and the race at 0d and increase the time frame between 0d and the recovering days at +1d, +2d, and +3d. A paragraph explaining this decision was added in the animals sections of methodology.

Results

Line 205: ‘showed statistically significant difference…’ with what? Among groups?

R: Yes, the statistical difference was between groups. This info was added in order to clarify the text.

Line 214: In the phagocytosis assay, there was a decrease … from group I, from 1 to 15. Sorry, I do not understand the sentence nor its correspondence in the table. Please clarify.

R: The sentence was rewritten. Group I was comprised from animals 1 to 15, as previously mentioned, and that what we were trying to say in the sentence. However, this information was excluded to avoid repetition.

Graphs

Change the spelling of all graph legends to English (e.g. phagocytosis vs. fagocytosis)

R: The corrections were done.

Graphic 3.2

Legend: A e B to be changed by A and B

R: The corrections were done.

Tables

Significance value should only be mentioned when significant

R: In accordance.

Table 3. 8 – Units are lacking

R: This table represents the cortisol concentrations variance between moments in group I. All values were inserted there.

Discussion

The authors suggest that the difference between the groups would be explained by their previous performance. We lack information regarding the current status of horses: were the horses at the same level of fitness at the time of sampling? Did the two groups participate in the same type of race? Were the two groups investigated in different seasons? Were their training sites similar (i.e. at the Jockey Club of São Paulo (JCSP))?

R: All horses were over two years old, with thirteen of them being Grand Prix winners, and. Even though all horses trained and were kept at the Jockey Club of São Paulo, there were slight differences when considering their training. All of them participated in the same race type in the spam of two months when the 0d samples were taken.

Many hypotheses are formulated in the discussion to explain the observed variations; several of them are supported by Pyne 1994, a reference that could not be retrieved from the reference list.

R: The reference was added.

Line 440 to 446: although not necessarily untrue, this hypothesis derived from a 40-year-old book chapter is too unspecific to be discussed here while respiratory function is not assessed in these groups of horses.

R: These sentences were delated, since there wasn’t a strong correlation with this study and the reference was old.

Line 447: it is difficult to relate respiratory infections or conditioning status to results as both have not been documented

R: Sentence deleted.

Line 498: we may wonder why you did not measured SAA

R: This suggestion is based on the analysis of the results from this study, in this way we could not perform SAA since samples were already processed for the other tests and it would be necessary to take other samples from all the different moments in order to measure SAA.

Remaining parts: again, they are more theoretical statements than hypotheses suggested by the results

R: The conclusion section was rewritten in accordance to the objective of the study.

Reviewer 3 Report

In this paper, the authors deal with the variation in the number and function of white blood cells in thoroughbred horses during competition or training.

The major critical point of this paper is the unfortunately non-homogeneous sampling due to the difficulty of being able to find blood samples in horse athletes during training or competitions. If it were possible, sampling should be implemented to make it more homogeneous.

Abstract

English needs to be revised for some spelling errors (e.g. essayes - assayes) and to make it more fluid and easy to understand

Introduction

The introduction section dealing with a well investigated topic must be implemented as a references.

Line 44 – e.g. Miglio, A., et al. Animals, 2021, 11(2), pp. 1–13, 447; Miglio, A. et al.Animals, 2020, 10(2), 317; Cappelli, K., et al.Veterinary Journal, 2013, 195(3), pp. 373–376

Line 61 and 68 – Add also more recent references

Deepen in the introduction section the already known effects of training and competition on white blood cells e.g. exercise stress lymphopenia and that known as the 'open window' theory that reflects a temporal association between intense exercise and increased susceptibility to opportunistic infection (e.g. paper from Karagianni A.E.).

Also with respect to cortisol in training and during the competition the introduction section must be implemented.

Material and Methods

This section needs to be expanded by describing sampling in more detail.

It is not clear how many samples are taken from the same subject, at what times, if during training, during the competition and when during the day.

An explanatory table with all the samples for each subject should be added, also adding all the other parameters that are monitored.

The table should indicate all the subjects to understand also how much consistency is for each age.

A two-year-old a foal is at first training and may respond differently than a horse of six-year-old. It would be important to indicate how long the animal has been competing for and how many seasons it has been in training.

The section “experiment design” section is better placed before the methodological part of the analyzes

Results

Results section is well written and organized even if the differences in withdrawals of the two groups are not intuitive to compare.

Discussion

In the discussion section the references are formatted differently than in the other sections of the text.

The discussion section is well structured and offers many important concepts in the adaptation to exercise.

Author Response

We would like to thank you for the considerations and advices. The response to all questions and specific comments was added below.

Comments and Suggestions for Authors:

In this paper, the authors deal with the variation in the number and function of white blood cells in thoroughbred horses during competition or training.

The major critical point of this paper is the unfortunately non-homogeneous sampling due to the difficulty of being able to find blood samples in horse athletes during training or competitions. If it were possible, sampling should be implemented to make it more homogeneous.

R: The JCSP has a policy that during the race week all selected horses can’t receive any invasive treatment, so it is not possible to take blood samples during this week. Furthermore, many samples from horses that weren’t enrolled in races were discarded by the end of the study, with the main inclusion criteria being that horses needed to take part in a race during the study conduction. The samples vary from horses of both sexes with age range of 2 to 6 years, being in their peak of the racing career, possessing prizes or not and being enrolled in Grand Prix races.

Abstract

English needs to be revised for some spelling errors (e.g. essayes - assayes) and to make it more fluid and easy to understand

R: Corrections done.

Introduction

The introduction section dealing with a well investigated topic must be implemented as a reference.

R: Introduction changed, and references added.

Line 44 – e.g. Miglio, A., et al. Animals, 2021, 11(2), pp. 1–13, 447; Miglio, A. et al.Animals, 2020, 10(2), 317; Cappelli, K., et al.Veterinary Journal, 2013, 195(3), pp. 373–376

R: References were added.

Line 61 and 68 – Add also more recent references

R: References were added.

Deepen in the introduction section the already known effects of training and competition on white blood cells e.g. exercise stress lymphopenia and that known as the 'open window' theory that reflects a temporal association between intense exercise and increased susceptibility to opportunistic infection (e.g. paper from Karagianni A.E.).

R: The paper by Karagianni A.E was added, and the introduction section was changed in order to better present the topic and introduce the discussed material and study objective and hypothesis.

Also with respect to cortisol in training and during the competition the introduction section must be implemented.

R: Section added.

Material and Methods

This section needs to be expanded by describing sampling in more detail.

R: The material and methods section was changed and a paragraph regarding the protocol was added.

It is not clear how many samples are taken from the same subject, at what times, if during training, during the competition and when during the day.

R: The number of samples taken from each horse varied between group 1 and group 2 animals. However, one sample was taken at each collection moment, with group 1 being collected at the moments -11d, -8d, 0d, +1d, +2d, +3d and group 2 -8, 0d, +1d and +5d. A phrase was added to the sample collection section in order to better clarify.

An explanatory table with all the samples for each subject should be added, also adding all the other parameters that are monitored.  The table should indicate all the subjects to understand also how much consistency is for each age.

R: The study didn’t analyze other parameters beside blood samples, since all animals were healthy and under veterinary care. We believe that the changes done to the manuscript are enough to clarify how many samples were taken from each horse. 

A two-year-old a foal is at first training and may respond differently than a horse of six-year-old. It would be important to indicate how long the animal has been competing for and how many seasons it has been in training.

R: Most animals in the study were 2 and 3 years old foals, with only one 6 years old horse.

The section “experiment design” section is better placed before the methodological part of the analyzes

R: Changed.

Results

Results section is well written and organized even if the differences in withdrawals of the two groups are not intuitive to compare.

R: Some tables and sentences were changed in order to clarify the article.

Discussion

In the discussion section the references are formatted differently than in the other sections of the text.

R: The references were changed to the journal format.

The discussion section is well structured and offers many important concepts in the adaptation to exercise.

R: Thank you!

Round 2

Reviewer 2 Report

I have no additional comment. 

Reviewer 3 Report

A two-year-old a foal is at first training and may respond differently than a horse of six-year-old. It would be important to indicate how long the animal has been competing for and how many seasons it has been in training.

R: Most animals in the study were 2 and 3 years old foals, with only one 6 years old horse.

Since you do not think it is necessary to insert a table, it is necessary that the age of the subjects +/- SD be written and I highly recommend, if only one horse is six years old and all the other 2 or 3 years old, to remove it as it is particularly uneven, inhomogeneous and to proceed with the analyzes without this sample.

Author Response

We would like to thank you for the review. A table has been added to section 2, containing all data about horses ages, sex and distance raced for collection moment 0d. 

Round 3

Reviewer 3 Report

As indicated by the first revision, the added table shows the inhomogeneous sampling which, in my opinion, was immediately indicated as a criticality of this work.
In the second review response it was said that only one horse was out of the average being 6 years old, when the table shows 5 five-year-old horses.
However, highlighting this weak point that will certainly affect the result, in my opinion the work can be published in this form.